# OpenReview forum: "Position: Stop Automating Peer Review Without Rigorous Evaluation"
_ICML.cc/2026/Position_Paper_Track — ICML 2026 Position Paper Track spotlight_

### Official Review · Reviewer_Eaj6 · 2026-03-12

**Significance:** 4
**Argument Clarity:** 3
**Rating:** 4
**Confidence:** 4

**Questions:**

Beyond embedding-based similarity, could you add an evaluation that more directly captures epistemic diversity? For example, cluster and annotate the weaknesses/concerns to compare (i) coverage of distinct critique points, (ii) complementarity/overlap across reviews, and (iii) key-defect hit/miss rates, to verify that higher textual similarity truly corresponds to fewer independent perspectives.
Robustness and external validity: Do the hivemind and laundering effects remain significant across different prompts, review pipelines (e.g., question-first vs. score-first), temperature/sampling settings, and multi-model reviewer ensembles? If the conclusions primarily apply to a particular configuration, the paper should state that boundary explicitly and provide a minimal robustness-check suite.
Is homogenization merely templated writing? Please recompute similarity after removing templated sections (e.g., standard Summary phrasing), or compute similarity only on weakness sentences, to test whether the hivemind conclusion still holds when boilerplate is controlled for.
Cross-area/track consistency: Given the broad topical range of ICLR submissions, please stratify the analysis by area or track (e.g., vision/NLP/theory) to determine whether hivemind is a general phenomenon or driven disproportionately by specific subfields.

**Alternative Views Section:**

Yes

**Compliance With Llm Reviewing Policy A Conservative:**

Affirmed.

**Discussion Potential:**

4

**Final Justification:**

The authors have added additional experiments and provided clarifications addressing my concerns. Based on these revisions, I maintain my original score.

**Paper Summary:**

This paper argues that before deploying general-purpose large language models to automatically generate peer-review reports and scores—a high-stakes component of the publication pipeline—we must establish a rigorous evaluation framework; otherwise, such automation may undermine the effectiveness and legitimacy of the review system. The authors propose two necessary conditions: (C1) automated reviewing should preserve reviewer diversity, avoiding a homogenized “hivemind”; and (C2) automated reviewing should be robust to gaming, so that decisions are not swayed by superficial changes unrelated to scientific merit. Using analyses of large-scale conference review data and simulated reviewer agents, the paper shows that AI-generated reviews can become more homogeneous and introduces the phenomenon of “paper laundering”, where a zero-shot rewrite of a manuscript can raise an AI reviewer’s score. Finally, the paper recommends pre-deployment adversarial robustness testing, accuracy validation, and transparent disclosure as safeguards.

**Position:**

Yes

**Position In Title:**

Yes

**Related Work:**

3

**Strengths And Weaknesses:**

Strengths:
Important, high-stakes topic: The paper frames “automated peer review” as a high-risk decision-making component that can directly influence publication outcomes and the allocation of scientific resources, and it stresses the urgency of conducting rigorous pre-deployment evaluation.
	Testable, operational claims: Rather than remaining at a purely normative level, the paper concretizes its concerns into two necessary conditions—preserving reviewer diversity and being robust to gaming—and presents experiments intended to demonstrate failure cases, thereby helping to set a clear agenda for follow-up research.

Weaknesses:
Misalignment between the C1 (review diversity) claim and its measurement: The paper primarily operationalizes “hivemind” via cosine similarity of full-review embeddings (IntraSim/InterSim). However, textual similarity can be strongly driven by templated writing, politeness boilerplate, and fixed review structure, and it does not directly measure the complementarity or disagreement in criticisms, defect discovery, or priority judgments. The paper also lacks annotation-based or qualitative evidence showing that AI reviewers truly “converge on the same set of flaws.”
The “real-world” comparison relies on third-party AI-review labels without quantifying labeling error: Conclusions drawn from grouping conference reviews into “fully AI-generated vs. others” are vulnerable to mislabeling noise. While the paper acknowledges uncertainty due to labeling and mixed human–AI usage, it does not provide manual spot checks or sensitivity analyses, weakening external validity and the robustness of the claims.
Limited coverage of models/prompts/strategies undermines a strong policy-style title conclusion: The key experiments focus on a small set of frontier models and a single, relatively constrained review prompt, without robustness checks across prompts, temperature/sampling settings, review pipelines (e.g., question-first vs. score-first), or multi-model configurations. As a result, the headline recommendation to “stop automating” reads more like a call for a research agenda than a conclusion firmly supported by broad, stable evidence.

**Support:**

2

---

> ### Author Rebuttal · Authors · 2026-03-30
>
> Thank you so much for this incredibly constructive and helpful review!
> We agree that our recommendation to stop automating peer review without rigorous evaluation resembles a call for a research agenda. This is part of the position we put forward and one of the main goals of our paper.
> We agree with your other points as well. We ran several new experiments addressing your specific suggestions, and the results consistently validate our initial findings.
>
> We provide all new results here: https://anonymous.4open.science/r/icml2026aireviewer/
>
> **Hivemind effect persists, and increases, when restricted to weaknesses and questions only (New Experiments 1 & 2).**
> We recomputed all similarity metrics after removing boilerplate, i.e., using only the weaknesses and questions sections, excluding summary and strengths.
> The hivemind effect  *increases* after this removal (New Experiment 1, Figure 1). Human IntraSim drops substantially without boilerplate (0.811 to 0.658), showing that humans write similar summaries but diverse critiques. AI IntraSim barely changes (0.882 to 0.835). The effect size rises from Cohen's d = 1.47 to d = 1.93. InterSim shows consistent patterns, with Claude's effect size increasing from d = 1.41 to d = 2.07. All p < 0.0001.
>
> We observe the same pattern in the wild (New Experiment 2, Figure 2): restricting 73,993 ICLR 2026 reviews to weaknesses and questions increases the effect size from d = 0.29 to d = 0.35. The hivemind effect reflects convergence in substantive critique, not shared templates.
>
> **Epistemic diversity evaluation.**
> While a full defect-level annotation study was infeasible within the rebuttal period, isolating weaknesses and questions sections serves as a direct proxy for epistemic content. These sections contain the substantive critiques and are free of the summary/strengths boilerplate that inflates stylistic similarity. The finding that the hivemind effect *strengthens* when restricted to these sections provides evidence that the convergence is in critique substance.
>
> **Hivemind effect is consistent across all subfields (New Experiment 3).** We stratified the in-the-wild hivemind analysis across all 21 ICLR 2026 primary areas (Figures 3 and 4). The effect is significant (p < 0.05) in every area, for both full reviews (New Experiment 3a, Cohen's d = 0.02-0.45) and weaknesses + questions only (New Experiment 3b, d = 0.06-0.53).  The hivemind effect occurs in all subfields, and is not driven by just some specific area.
>
> **Pangram label accuracy validated by independent human judgments (New Experiment 4).**
> Crucially, the labels we use to identify AI-generated reviews in the wild are from Pangram (Emi, 2025).
> Jabarian and Imas (2025) found that Pangram achieves "near-zero FNR and FPR rates" and that Pangram can "satisfy a strict cap (FPR ≤ 0.005) without sacrificing accuracy".
> In their comparison against ICLR 2022 reviews, Pangram reported 0 false positives (https://x.com/max_spero_/status/1989923081993809973?s=20).
> To further quantify Pangram's precision, we performed New Experiment 4 using human author judgments. We systematically searched all 159,775 author comments across ICLR 2026 for complaints about AI-generated reviews, identifying 58 genuine accusations (manually verified). Of these, 50 (86.2%) were independently flagged by Pangram as fully AI-generated; only 2 (3.4%) were classified as fully human-written. Notice that accusations targeting Pangram-flagged reviews often provided concrete evidence (e.g., hallucinated citations, one explicit admission of ChatGPT use), while accusations targeting non-flagged reviews were vague. While author accusations are not a perfect ground truth, authors know their work best and have likely read the reviews in detail, making this a reasonable validation proxy.
>
>
> **Paper laundering is robust across prompts, models, and reviewers (New Experiment 5).**
> We expand robustness checks as follows: test the robustness of paper laundering across 4 prompts (two new additional simple paraphrases and one prompt that adds a sentence instructing the model that the paper should be rewritten so that it jailbreaks the AI reviewer), 2 launderer models (GPT-5.1, GPT-5.4), and 3 reviewer models (GPT-5.1, GPT-5.4, Claude-sonnet-4.6), which yields 24 conditions. The effect is significant in almost every condition (see New Experiment 5a, Figure 6 and New Experiment 5b, Figure 7).
> We deliberately rely on the review pipeline proposed and validated by Bianchi et al. (2025b) rather than testing experimental or untested pipelines.
>
> New References:
> - Brian Jabarian and Alex Imas, "Artificial Writing and Automated Detection," NBER Working Paper 34223 (2025), https://doi.org/10.3386/w34223.

---

> > ### Author Rebuttal · Reviewer_Eaj6 · 2026-04-03
> >
> > The authors have added additional experiments and provided clarifications addressing my concerns. Based on these revisions, I maintain my original score.

---

### Official Review · Reviewer_n2db · 2026-03-12

**Significance:** 4
**Argument Clarity:** 3
**Rating:** 5
**Confidence:** 3

**Questions:**

See weaknesses above.

**Alternative Views Section:**

Yes

**Compliance With Llm Reviewing Policy A Conservative:**

Affirmed.

**Discussion Potential:**

3

**Final Justification:**

Thanks for the detailed response, which addressed my concern. Overall, I think this paper points out a serious problem of people blindly using LLMs. Therefore, i am glad to increase my score from 4 to 5.

**Paper Summary:**

This paper claims that today’s AI systems should not be used to produce paper reviews. After analyzing the empirical comparison of human and AI generated ICLR 2026 reviews, the authors identify two critical issues, including the hivemind effect and paper laundering. Alternative views and call-to-actions are provided for critical thinking.

**Position:**

Yes

**Position In Title:**

Yes

**Related Work:**

3

**Strengths And Weaknesses:**

Strengths:
- The paper is well-written;
- The review issue (especially regarding the use of LLMs) is clearly important nowadays;
- The analysis is comprehensive and in-depth.

Weaknesses:
- A surprising point I learned from this paper is that with very low or even no cost people can improve their paper. If we purely see the fact, this can be beneficial for some people. However, this paper uses the term "paper laundering", which sounds a bit negative. If after polishing the paper using AI systems, the quality of paper really improves, then it should be a positive thing. Then the question comes to whether the quality of paper truly improves with the use of AI systems. Maybe the authors should incorporate some metrics to check in more details.

**Support:**

4

---

> ### Author Rebuttal · Authors · 2026-03-30
>
> Thank you for the positive feedback! We agree that AI is valuable for improving paper clarity, and we do not argue against AI-assisted writing. The distinction we draw is between human-supervised polishing and fully automated rewriting without any oversight.
>
> While we acknowledge that some laundering edits may improve readability, our manual inspection (Appendix E.1) shows that fully automated zero-shot rewrites frequently cross the line from polishing into hallucination: examples include fabricated ablation sections with invented accuracy numbers, unproved theorems, and generic "answers to reviewer questions", none of which is grounded in actual experiments. For this reason, we believe that paper laundering, as we study it, poses serious risks.
>
> More fundamentally, our contribution is showing that AI reviewers are trivially gameable. We use a handful of zero-shot prompts without any optimization, yet the scores are significantly increased (see our new Experiment 5, in particular Figures 6 and 7 here: https://anonymous.4open.science/r/icml2026aireviewer/). This is a concrete instance of Goodhart's Law, i.e., when AI review scores become the target, authors can optimize for the metric rather than the substance. A review system where scores can be inflated without genuine scientific improvement is not fit for making consequential acceptance decisions in full automation.

---

> > ### Author Rebuttal · Reviewer_n2db · 2026-04-02
> >
> > Thanks for the detailed response, which addressed my concern. Overall, I think this paper points out a serious problem in using LLMs. Therefore, i am glad to increase my score from 4 to 5.

---

### Official Review · Reviewer_pbBH · 2026-03-13

**Significance:** 3
**Argument Clarity:** 4
**Rating:** 5
**Confidence:** 4

**Questions:**

Why are the absolute scores in Fig 2 so much higher (0.8+) than the ones in Fig 1, even though they’re measuring inter-review similarity with the same embedding model?

**Alternative Views Section:**

Yes

**Compliance With Llm Reviewing Policy A Conservative:**

Affirmed.

**Discussion Potential:**

4

**Final Justification:**

This paper presents a fairly well-supported case on an important issue, arguing why we should wait for more data before adopting LLM use in scientific peer review.  While I still have a few questions, the author rebuttal (which included compelling new experiments) was important in raising my score from a weak accept to an accept.

**Paper Summary:**

This paper argues that we should “hit the brakes” on AI reviewing until we can establish that AI reviewers are accurate and not gamable.  As part of its argument, it presents new experimental results showing that AI reviews lack diversity (what the paper calls a “hivemind” effect) and are gamable with superficial edits that do not improve the content.

**Position:**

Yes

**Position In Title:**

Yes

**Related Work:**

4

**Strengths And Weaknesses:**

Strengths

In general, this paper presents a well-researched, fact-based argument for an important issue that is pressing for the community.  The arguments and the experimental results are presented clearly, and draw on relevant literature for support.  I found the discussion of alternative views to be helpful as well, identifying important counterarguments.

The paper investigates whether the “AI laundering” improves the papers (it doesn’t), which was a helpful verification.

The paper includes specific, constructive steps that the community could take to pave the way toward automated reviewing (Sec 6).

Weaknesses

The hivemind effect in Fig 1 seems quite weak (difference in means in Fig 1 seems small relative to each distribution’s standard deviation—I realize the difference is statistically significant, but is it practically significant?).  Further, one possible explanation of the greater similarity of AI-written reviews is because of stylistic, rather than content effects.  The hivemind effect only damages peer review quality if the content, not merely the style, of the reviews is inappropriately homogeneous—but the paper’s experiment does not tease apart the two factors.  Fig 2 is more convincing, although I wonder if this is due only to the specific choice of AI reviewer used there.

The particular experimental results presented here, hivemind and laundering weaknesses of AI reviewers, are interesting and potentially important, although neither are among the most significant concerns with AI reviewers in my assessment.  More concerning would be if the AIs are inaccurate at scoring (in particular, less accurate than humans), superficial, or miss important flaws.  The paper discusses a few prior works that touch on some of these other concerns.  One particular aspect I found less convincing is that the algorithmic monoculture argument here is only stated as a theoretical weakness; it seems like if that were indeed a practical concern, one ought to be able to show it today because the average of multiple AI reviews would be less accurate than the average of human reviews (because of the correlated errors made by the AIs).

**Support:**

3

---

> ### Author Rebuttal · Authors · 2026-03-30
>
> Starting with the question about Fig. 1 vs. Fig. 2 absolute scores: Fig. 2 represents simulated reviews using a single AI reviewer agent with a uniform prompt, whereas Fig. 1 reflects in-the-wild reviews, where reviewers used diverse prompts, models, and hybrid human-AI workflows.
>
> Thank you for the helpful comments! We have addressed those with a set of new experiments, which you can find here: https://anonymous.4open.science/r/icml2026aireviewer/.
>
> These new results consistently validate our initial findings:
>
> **Homogeneity extends to weaknesses and critiques, not just boilerplate paper summary (New Experiments 1 & 2).**
> We recomputed all similarity metrics using only the weaknesses and questions sections, excluding summary and strengths to control for stylistic boilerplate. The effect sizes *increase*: IntraSim Cohen's d rises from 1.47 to 1.93; in-the-wild InterSim d rises from 0.29 to 0.35 (New Experiment 1, Figure 1; New Experiment 2, Figure 2). Human reviews become much less similar without boilerplate (IntraSim: 0.811 to 0.658), while AI reviews barely change (0.882 to 0.835). The homogeneity extends to substantive critique, not just summary and strengths.
>
> **Algorithmic monoculture has measurable practical consequences (New Experiment 6).**
> We agree that demonstrating practical consequences of algorithmic monoculture is important. In New Experiment 6 (Table 1), we show that averaged **AI scores** are significantly worse predictors of the final ICLR 2026 acceptance decisions than averaged human review scores.
> On the matched subset of 8,015 papers with both human and AI reviews, averaged human scores predict acceptance with AUC = 0.822, while averaged AI scores achieve only AUC = 0.710 (non-overlapping 95% CIs). We find the same pattern with stricter matching (≥2 reviews each: human AUC = 0.798 vs. AI AUC = 0.751).
>
>
> **In-the-wild effect size is a conservative lower bound.**
> The in-the-wild effect (Cohen's d = 0.29) is conservative for two reasons. First, the "other" group likely contains AI-assisted reviews that Pangram did not flag as fully AI-generated, attenuating the contrast. Second, restricting to weaknesses and questions increases the effect to d = 0.35 (New Experiment 2, Figure 2), and in New Experiment 3b, the effect reaches d = 0.53 in some areas (Figure 4).
>
>
>
> **Hivemind is not specific to one AI reviewer model (New Experiment 3).** The in-the-wild hivemind effect is significant across all 21 ICLR primary areas (Figures 3 and 4), where reviewers presumably used diverse models and prompts. In our simulations, both GPT-5.1 and Claude show the effect.

---

> > ### Author Rebuttal · Reviewer_pbBH · 2026-04-01
> >
> > Thank you for the thoughtful response.  I feel like stylistic factors can still exist and affect the homogeneity measurement even within weaknesses and critiques---a manual analysis or even LLM as judge rather than embedding similarity could address that concern.  I am actually not quite sure why summary and (especially) strengths would have more boilerplate necessarily.
> >
> > The algorithmic monoculture result is quite interesting---I worried a little bit about the fact that these reviews are actually inputs to the accept/reject decision you're treating as the dependent variable.  I couldn't really think of a reason why that would break the experiment, but I'd encourage the authors to think about this and discuss it in their revisions.
> >
> > Given the new experiments and answers to my questions, I will raise my overall score to a 5.

---

### Official Review · Reviewer_ufzq · 2026-03-13

**Significance:** 4
**Argument Clarity:** 4
**Rating:** 5
**Confidence:** 4

**Questions:**

For the high correlation among AI reviewer scores than among human reviewers, do you consider the quality of human reviews? While reviews generated by Claude / GPTs are at the same professional level, human reviewers have diverse backgrounds, expertise, and review criteria, making them more diverse but may not reflect the true quality of the paper.

Do you think the review diversity and gameability can be mitigated if multiple AI reviewers (e.g., different LLMs) are used for providing the same review?

**Alternative Views Section:**

Yes

**Compliance With Llm Reviewing Policy A Conservative:**

Affirmed.

**Discussion Potential:**

4

**Final Justification:**

My concerns have been adequately addressed.

**Paper Summary:**

This paper focuses on the automatic peer review in current scientific conferences and identifies two key issues: the loss of reviewer diversity and vulnerability to gaming. Therefore, this paper presents the position that the current review process should not use automatic peer reviewing, and we need to address these issues with a specific technology instead of the general-purpose LLMs without rigorous evaluation.
The authors design experiments on 75,800 reviews and find that AI reviews show significant within-group similarity and agreement compared to other human reviews, and they are more similar across different papers. Furthermore, they find that paper laundering can significantly improve AI-generated review scores, making it vulnerable to such an attack

**Position:**

Yes

**Position In Title:**

Yes

**Related Work:**

4

**Strengths And Weaknesses:**

Strengths
* This paper focuses on a very important and highly controversial problem.  As more and more reviewers use LLMs to generate reviews, it is important to investigate the potential risk behind it, which can raise discussion and attention in the AI community.
*  The authors provide solid empirical support for their position by AI reviewer hivemind effect and paper laundering. They conduct analysis on a large scale of reviews and show that the AI reviews lack diversity, as indicated by a higher agreement and similarity across papers.
* This paper also conducts gaming between paper writing and AI reviewing and finds that AI reviews are more vulnerable to gaming with the simple zero-shot rewriting. This further causes the intellectual monoculture where the laundered papers are more similar, highlighting the serious harm.

Weaknesses
* The accuracy and reliability of the AI-generated label in the review corpus are not clear. The main experiment uses the labels from Emi (2025) to annotate the AI-generated reviews. Is it possible that these reviews are identified as AI-generated because they share some patterns that can be identified by Pangram? In other words, is it possible that there are some diverse AI reviews, but they are mistakenly labelled as human reviews?
* From Table 4, we can find that the main reason is that LLMs use similar template phrases to ask questions or summarize content, instead of asking the same questions or directly reusing comments. Such behaviors, while they do harm the diversity of the review surface text, may not necessarily harm the quality of the reviews. Human can also choose to reuse their preferred phrases or templates to express their different opinions.

**Support:**

3

---

> ### Author Rebuttal · Authors · 2026-03-30
>
> Thank you for the thoughtful and constructive feedback!
> We provide a set of new experimental results here, addressing your specific suggestions: https://anonymous.4open.science/r/icml2026aireviewer/
>
> These new results consistently validate our initial findings:
>
>
> **Pangram labels are reliable (New Experiment 4).** Thank you for raising this important concern. The labels we use come from Pangram (Emi, 2025).
> Jabarian and Imas (2025) found that Pangram achieves "near-zero FNR and FPR rates" and that Pangram can "satisfy a strict cap (FPR ≤ 0.005) without sacrificing accuracy".
> In their comparison against ICLR 2022 reviews, Pangram reported 0 false positives (https://x.com/max_spero_/status/1989923081993809973?s=20).
> To further validate these labels, we performed New Experiment 4 (Figure 5) using human author judgments. We systematically searched all 159,775 author comments across ICLR 2026 for complaints about AI-generated reviews, identifying 58 genuine accusations (manually verified). Of these, 50 (86.2%) were independently flagged by Pangram as fully AI-generated; only 2 (3.4%) were classified as fully human-written. Notice that accusations targeting Pangram-flagged reviews often provided concrete evidence (e.g., hallucinated citations, one explicit admission of ChatGPT use), while accusations targeting non-flagged reviews were vague. While author accusations are not a perfect ground truth, they provide further evidence that the labels we use are of high quality.
>
>
> **AI critique convergence goes beyond template summary and strength phrases (New Experiments 1 & 2).**
> We agree that similar template phrasing alone does not necessarily harm review quality. To test whether the homogeneity extends beyond surface-level templates, we recomputed all similarity metrics using only weaknesses and questions, excluding summary and strengths (new experiment 1, see Figure 1).
>
>
> The effect sizes increase after removing boilerplate. Human reviews become much less similar without templated sections (IntraSim: 0.811 to 0.658), while AI reviews remain nearly as homogeneous (0.882 to 0.835). This holds both in simulation and in the wild (new experiment 2, see Figure 2, Cohen's d: 0.29 to 0.35). AI reviewers are not just reusing preferred phrases to summarize the paper, their critiques converge.
>
>
> **Human reviews better reflect paper quality (New Experiment 6).** Regarding whether AI review agreement reflects true paper quality better than diverse human reviews: in New Experiment 6 (Table 1), we show that averaged human review scores are substantially better predictors of final ICLR 2026 acceptance decisions than averaged AI scores (AUC = 0.822 vs. 0.710 on matched papers, non-overlapping 95% CIs). Please see our response to Reviewer pbBH for full details. The higher agreement among AI reviewers does not translate into more accurate assessment of paper quality.
>
>
> **Multiple AI reviewer models do not eliminate gameability (New Experiment 5).**
> Across 3 different reviewer models (GPT-5.1, GPT-5.4, claude-sonnet-4.6), 2 launderer models, and 4 prompts, the laundering effect is significant almost all of the 24 conditions (see New Experiment 5a, Figure 6 and New Experiment 5b, Figure 7). Using different LLMs as reviewers does not eliminate the vulnerability.
>
> New References:
> - Brian Jabarian and Alex Imas, "Artificial Writing and Automated Detection," NBER Working Paper 34223 (2025), https://doi.org/10.3386/w34223.

---

> > ### Author Rebuttal · Reviewer_ufzq · 2026-04-02
> >
> > My concerns have been solved, and I would like to keep my original score of 5 as it already suggests acceptance.

---

### Decision · Program_Chairs · 2026-04-30

**Decision:**

Accept (spotlight)

**Comment:**

This is a good position paper on a timely and high-stakes issue. Its main contribution is that it does not stop at saying “AI review is risky,” but instead grounds that claim in concrete empirical failure modes: homogeneity across reviews and vulnerability to superficial gaming. The reviewers were overall very positive, and the main concerns were addressed well in rebuttal with additional experiments, including checks beyond boilerplate text, robustness across settings, and further validation of the review labeling pipeline.

I also appreciate that the paper does not overclaim a complete solution. Instead, it argues for a more careful scientific basis before deploying automated reviewing in consequential settings. That feels like an appropriate and useful contribution for the position track.